# Characterization of *Polylepis tarapacana* Life Forms in the Highest-Elevation Altiplano in South America: Influence of the Topography, Climate and Human Uses

**DOI:** 10.3390/plants12091806

**Published:** 2023-04-28

**Authors:** Victoria Lien López, Lucia Bottan, Guillermo Martínez Pastur, María Vanessa Lencinas, Griet An Erica Cuyckens, Juan Manuel Cellini

**Affiliations:** 1Laboratorio de Investigaciones en Maderas (LIMAD), Facultad de Ciencias Agrarias y Forestales (FCAyF), Universidad Nacional de La Plata (UNLP), La Plata 1900, Argentina; jmc@agro.unlp.edu.ar; 2CCT-La Plata (CONICET), La Plata 1900, Argentina; 3Centro Austral de Investigaciones Científicas (CADIC), Consejo Nacional de Investigaciones Científicas y Te\u0301cnicas (CONICET), Houssay 200, Ushuaia 9410, Argentina; lucia.bottan@agro.unlp.edu.ar (L.B.);; 4Instituto de Ecorregiones Andinas (INECOA-CONICET-UNJu) y Centro de Estudios Territoriales Ambientales y Sociales (CETAS-UNJu), Alberdi 47, San Salvador de Jujuy 4600, Argentina

**Keywords:** tree growth form, stem, multi-stemmed, scrublands, scrub, tree, Argentinean highlands, high Andean vegetation

## Abstract

In the upper vegetation limit of the Andes, trees change to shrub forms or other life forms, such as low scrubs. The diversity of life forms decreases with elevation; tree life forms generally decrease, and communities of shrubs and herbs increase in the Andean highlands. Most of treeline populations in the northwestern Argentina Altiplano are monospecific stands of *Polylepis tarapacana*, a cold-tolerant evergreen species that is able to withstand harsh climatic conditions under different life forms. There are no studies for *P. tarapacana* that analyze life forms across environmental and human impact gradients relating them with environmental factors. This study aims to determine the influence of topographic, climatic, geographic and proxies to human uses on the occurrence of life forms in *P. tarapacana* trees. We worked with 70 plots, and a new proposal of tree life form classification was presented for *P. tarapacana* (arborescent, dwarf trees, shrubs and brousse tigrée). We describe the forest biometry of each life form and evaluate the frequency of these life forms in relation to the environmental factors and human uses. The results show a consistency in the changes in the different life forms across the studied environmental gradients, where the main changes were related to elevation, slope and temperature.

## 1. Introduction

Life forms are functional types that have been used to describe the adaptation of vegetative structures of plants to certain ecological conditions. They refer to the general physiognomy of the tree or the common habit of the individual [1,2], and are used to interpret the functionality of vegetation [3,4] as well as to group individuals with similar morphologies [5], which allows comparisons between floras from different regions [4]. Solbrig [6] described life forms as functional groups based on a single character and Raunkiær expressed that, in response to harsh environments, plants can develop adaptations that allow the protection of renewing shoots [7].

In the upper limit of vegetation, trees change their growth strategy to shrub or other life forms, such as low scrubs [8]. These ecosystems are characterized by a marked thermal seasonality defined by long winters and short growing seasons, which greatly influence plant growth. Trees are subjected to the synergistic actions of stress, due to extreme fluctuations in temperature, drought, nutrient limitations and high levels of radiation [9,10]. Tree distribution boundaries are rarely sharp, and the transition from tree to shrub-only stages may be fragmented and stretched over a few meters in steep terrains or over many kilometers in flat terrains [8].

The topography, soils and the degree of human disturbance can modify the distribution and structure of arid and semi-arid vegetation [11]. The diversity of life forms and tree growth forms decreases with elevation; tree life forms generally decrease while shrubs and herbs increase in the Andean highlands [4]. Elevation represents the best example of an environmental gradient, where resources change due to a complex combination of climatic factors (e.g., temperature, rainfall, soils and substrate stability) [12], and it is a decisive factor that shapes the spatial patterns of plant life forms [13]. Therefore, the identification and estimation of different tree life forms are relevant for the evaluation of ecosystem structure and function [14,15].

In the Andean regions at the tree line, trees show a series of morphological and physiological responses to face extreme low temperatures and other environmental stresses, such as drought (caused by greater levels of evaporation), lower rainfall and higher solar radiation [16,17]. The response to this type of environmental stress is a reduction in the aboveground biomass, manifested as a decrease in tree height [18]. In this area, some individuals with unbranched or slightly branched woody stems have different tree life forms [19], standing out in places with little vegetation that does not exceed 50 cm in height [20]. These observations have generated greater interest in the adaptive importance of tree forms in the highlands [21]. In the Andean forests, there are individuals that growth in prostate or creeping forms, due to the effect of wind and snow, generating crooked trunks [19]. This type of growth is described for *Nothofagus pumilio* (Poepp. *et* Endl.) Krasser and *Austrocedrus chilensis* (D. Don) Pic. Serm. and Bizzarri, where low temperatures and drought are the limiting factors that condition the growth and tree development [22], as well as for *Polylepis tarapacana* Phil. forests in the Chilean highlands [19].

In the highlands of the South American Andes, *P. tarapacana* forests present differences in forest structure at different elevations and climate [23]. This species is sensitive to soil moisture content and, in response to drought, it can distribute its biomass in multiple individual stems with smaller diameters and heights to conserve the available moisture of the plant [19,21].

Rios [24] and Saavedra [19] described two life forms for *P. tarapacana* in Chilean forests: (i) single stem trees and (ii) multi-stemmed trees. These authors calculated the frequency of each tree life form, but they did not assess its relationship with environmental gradients or past human uses. Due to the correlation between the environment and the structure of these forests [21,23], it is expected that there is some influence of environmental factors on tree life forms [13]. Therefore, there is a need to review the classification developed by Ríos [24] and Saavedra [19], and propose a broader classification of tree life forms based on those described for different tree-line life forms [25,26,27,28,29,30,31,32].

Tree life form classifications, which can better present the tree line [26], are: (i) “*Treelets*”, or trees shorter than 5–10 m, and (ii) certain scrubby but very tree-like forms of 1–5 m, called “dwarf-trees”. The noun ‘arborescent’ is used to refer to woody plants that branch near the ground level. (iii) “*Arborescents*” are thus an intermediate form between trees and shrubs. (iv) “*Krummholz*” is a wind-stunted woody scrub occurring primarily at the tree-line and other exposed sites in mountains. (v) “*Shrubs*” are woody plants with multiple stems arising from the ground level. (vi) “*Dwarf shrubs*” are woody, generally small-leaved chamaephytes, not taller than 0.50 m, and more commonly 0.30 m. (vii) “*Cushion shrubs*” are small krummholz forms occurring near to constantly desiccating, often cold winds, with a completely limited vertical branch extension, resulting in an extremely dense mass of short branches forming a flat or rounded shape. Likewise, (viii) “*brousse tigrée*” (tiger bush) appear on slope gradient bands of vegetation formed perpendicular to this direction [26,29,30,31,32].

There are no studies that relate *P. tarapacana* life forms with environmental and human factors along elevation gradients [33], which is a topic of great ecological importance as a species adapted to very restrict environmental conditions. Moreover, it is considered a Near Threatened species [34] due to human impact, which mainly led to habitat degradation caused especially by the extraction of wood for fuel and construction in the area [35]. Some studies have attempted to make comparisons among plant life forms in the Altiplano region, but they always study different species in a community [13,36] and are not focused on a single species that might present different life forms. These studies are needed to assess the variability in tree life form composition within the region. There are also some intriguing differences in this region that deserve attention. For example, where are the tree life forms with only one trunk found? Is it more frequent to find multi-stemmed individuals? Does the extraction and use of firewood and poles modify the frequency of single-trunk life forms in the landscape? This and other similar questions are based on more or less anecdotal evidence of distribution patterns, which generally relate to forest structure data rather than form description.

In this paper, we elucidate how tree life forms change along different environmental and human use gradients. This study aims to determine the influence of topographic, climatic, geographic and human use factors on the occurrence of *P. tarapacana* life forms. In particular, we aim to answer the following questions: (i) Which tree life forms does this species have and what tree-specific biometric characteristics do the different tree life-forms have? (ii) How does the tree life forms and its distribution change with topography (elevation, slope and aspect), climate (temperature and precipitation), life zones and proxies of human uses (human footprint and distance to towns)? 

## 2. Results

The 70 plots presented a large heterogeneity in topography (e.g., location, elevation, slope and aspect), where the N–S range was 156 km and W–E was 61 km. The elevation gradient was 789.1 m, where four life zones occur: Tropical alpine moist tundra (TAMT), Tropical alpine wet tundra (TAWT), Tropical subalpine dry scrub (TSDS) and Tropical subalpine moist forest (TSMF). Likewise, the average distance to towns was 9.7 ± 4.7 km (average ± standard deviation). During the surveys, 1801 trees were recorded, with an average area of 1.29 m^2^ of tree crown, 7.8 cm of DBT and 86.1 cm of H.

### 2.1. Life Forms of P. tarapacana

We found four different life forms in the studied plots: arborescent (Ar), dwarf trees (Dt), shrubs (Sh) and brousse tigrée (Bt). The Ar life form has a unique base and branches that cover the entirety of the trunk, protecting buds from wind damage on the bark (Figure 1). The Ar life form presents a highly variable size between individuals (DAB 8.3 ± 7.0 cm, H 73.8 ± 51.6 cm). Its tree crown was 0.42 ± 0.57 m^2^ and showed a CsR circular to oval shape (1.3 ± 0.3). In Dt, the crown is observed at the end of the main axis of the trunk, which is not covered by branches and has frequent signs of crown dieback (Figure 1). The DAB was 10.9 ± 7.4 cm and H was 101.0 ± 55.3 cm. Its TC and CsR were similar to those of Ar, with values of 0.35 ± 0.41 m^2^ for TC and a similar CsR (1.3 ± 0.3). In the shrub life form, we observed multiple trunks, where the shortest ones were located at the periphery of the bush and the tallest ones in the center. A high mortality of the central trunks was observed in some shrub life forms (Figure 1). The size was 9.1 ± 6.3 cm for DAB and 100.7 ± 49.4 cm for H. A broad and highly variable tree crown (size and shape) was observed, measuring 2.0 ± 2.4 m^2^, while the CsR value was similar to that of Ar and Dt (1.3 ± 0.3) (Figure 1). Brousse tigrée (Bt) occurs in *P. tarapacana* in bands, with different sizes of trunks, where the smallest were in the direction of the slope and the tallest were upslope (Figure 1). In the brousse tigree life form, some trunks located upslope were dead or with a low presence of live leaves. The size was 4.3 ± 3.0 cm for DAB and 57.6 ± 27.1 cm for H, where the tree crown was 1.4 ± 1.6 m^2^ with an elongated shape (2.74 ± 1.36).

The size of the individuals among the different life forms presented significant differences (DAB H_Test_: 77.1, *p* < 0.0001, and H = H_Test_: 94.8, *p* < 0.0001), where the smallest values were observed in Bt, followed by Ar, Sh and finally Dt (Figure 2). A higher frequency (H_Test_: 305.0, *p* < 0.0001) of small crowns was observed in Ar and Dt, intermediate values were observed in Bt, and the highest values were observed in Sh. The crown spread ratio did not present differences among Ar, Dt and Sh, while Bt was always more elongated, with an average value of 2.74 and a maximum value of 9.33 (H_Test_: 185.6, *p* < 0.0001). The vitality of the individuals, classified by life form, presented significant differences (H_Test_: 41.3, *p* < 0.0001), following the gradient of Dt < Sh < Ar = Bt, with Dt values close to intermediate vitality and Ar and Bt close to healthy. We provide complementary information of the biometric characteristics of each life form in Table A1 in Appendix A.

### 2.2. Changes in the Frequency of Life Forms According to Topographic, Climatic, Geographic and Human Use Factors

The frequency of each of the four life forms was analyzed by plot, and showed a higher value for Sh (50%), followed by Ar (24%), Bt (16%) and Dt (9%). Table 1 shows that topographic factors (elevation, slope and aspect) significantly affect the frequency of the defined life forms. There were significant differences for all *P. tarapacana* life form frequency in elevation, while slope only showing differences in Ar and Bt. At higher elevations, the frequency of Ar and Dt was higher (>30% in Ar and >13% in Dt for >4700 m a.s.l.), while for Sh and Bt, the frequency decreased (from 59% in <4400 m a.s.l. to 40 % at >4700 m a.s.l. in Sh and from 19% to 12% at the same elevations for Bt). In the same way, there is an inverse relationship between the slope and frequency of Ar, and a direct one in Bt. For the different aspects (north and east aspects), we did not detect significant differences among the studied life forms.

Differences in the frequency of *P. tarapacana* life forms were observed in relation to the following factors: life zone, annual mean temperature and distance to towns (Table 2). AMT showed a similar pattern to those observed in the elevation gradient analysis (Table 1), where the frequencies of the lowest temperatures are those located at the higher elevations. Additionally, the frequency of all life forms presented significant differences among the different life zones, with a greater number of Ar and Dt in the alpine life zone (TAMT and TAWT) and Sh and Bt in the subalpine areas (TSMF and TSDS). The alpine life zones are located at higher elevations compared with the subalpine zones, which corroborates the change in the structure described in the elevation analysis. For temperature, it was observed that the frequency of Ar and Dt increased in colder zones, and the opposite trend occurred for Sh. Likewise, neither AP nor HF showed significant differences. However, it was observed that the frequency of Ar and Bt increased and decreased, respectively, with the distance to towns (DTT).

Topographic (elevation, slope and aspects), climatic (temperature and precipitation), and human use (human footprint and distance to towns) factors were used to define the frequency of life forms per plot (Figure 3). The plots were categorized into one simple (Sh n = 12), three doubles (Sh|Ar n = 22, Sh|Bt n = 18 and Sh|Dt n = 4), one triple (Ar|Bt|Dt n= 5) and one multiple (M n = 9) in relation to the abundance of life forms. Elevation and slope presented two differentiated groups, where, in the plots with multiple forms, Sh|Ar and Sh|Dt occurred in high- and low-slope areas, shrubs occurred at low-elevation and -slope areas, and finally the formations with Bt occurred on high slopes. The aspect factor presented less marked differences, with one single group located in the north aspect and dwarf trees (Dt) occurred in the west aspect, while shrubs with Bt and Ar tended to occupy the east aspects. Regarding human footprint (HF) and DTT, the Dt life form was observed far away from towns, while shrubs occurred closer to towns. The HF showed a gradient in which Dt and Ar were found in values close to 0, and as this value increased, Sh and Bt were more frequently observed.

## 3. Discussion

In this study, a new proposal to classify life forms of *P. tarapacana* was presented. The new life forms considered were arborescent, dwarf trees, shrubs and brousse tigrée. This proposal differs from that of Ríos [24] and Saavedra [19] as it divides the single stem category into two (Ar and Dt) and the multi-stem into another two (Sh and Bt). The justification for this new classification lies in the marked differences in their specific biometric characteristics as well as in the described influence of topographic, climatic and human uses factors, mainly on their frequency. The inclusion of the additional life form categories (Ar and Bt, novel for this species) help to achieve a more comprehensive overview of the life form composition for *P. tarapacana.*

### 3.1. Frequency of P. tarapacana Life Forms in the Argentinean Altiplano

Several authors agree in the classification of the life forms of *P. tarapacana* into single trunk and multi-trunk [19,24], concluding that the highest proportion of individuals found correspond to multi-trunk. These observations were made in small and specific areas of the distribution of *P. tarapacana*, and the present work offered results in a larger study area, covering most of the heterogeneity of its distribution (e.g., location, elevation, slope and aspects).

According to Mooney [37], there should be, for a given combination of climate and community succession, an optimal dominant life form. With this, a high degree of similarity in life forms would be expected at each of the sampled stands in this study because a large environment condition was covered. We found that the highest frequency of life form corresponded to Sh (50.3%), where the highest proportion was related to the extreme climatic conditions of the study area. The Altiplano plateau of the central Andes represents one of the harshest places on Earth for plant growth [8], where shrubs are associated with disturbed and stressful environments [38,39], and due to their lower height, plants take advantage of the relatively favorable climate near the surface [40,41]. A relatively smooth surface reduces the turbulence and, thus, minimizes heat loss, implying that, in calm and clear conditions, the daytime temperature of shrubs is generally higher than the air temperature [42]. The highest proportion of Sh was found by other authors in the Chilean *P. tarapacana* [19], where 64% of the individuals have a multi-stem tree life form and only 36% present a single stem. Ríos [24] found, further south in the Province of Iquique (Chile), that the proportion of life forms changes according to different topographic conditions. Likewise to other species that inhabit extreme ecosystems [43], the different life forms of *P. tarapacana* show a great phenotypic plasticity (i.e., an ability to respond to changes in their architecture to adapt to different environmental variations throughout their distribution).

### 3.2. Biometric Characteristics of Different P. tarapacana Life Forms

The size and shape of the life forms was determined by the growth restrictions that affect the lateral shoots and apical buds as well as the ability of plants to produce shoots from the root [4,7,13]. For example, in our study, Shrubs (Sh) presented the largest tree crown. Epicormic shoots under the bark facilitate the horizontal growth of shrubs [44], so they can expand horizontally as their length and mass increase to capture more light than a small tree that tends to grow mainly upwards [45,46,47]. In cold and alpine areas, low vegetation survives extreme weather and strong winds better, due to a better aerodynamic resistance [48,49,50]. In *P. tarapacana* shrubs, the highest trunk mortality occurs in the central zone due to the risk of cavitation caused by drought and freezing [51,52]. This could be a beneficial adaptation to hostile environments, such as the highlands, where the maximum height of the trees is determined in part by the problem of exposure to water in the upper crown. A lack of water can cause xylem embolism [53] and the risk of cavitation increases with stem height due to gravity. Related to this, both the brousse tigrée (Bt) and shrub (Sh) life forms, by developing multiple stems, have a continuous horizontal growth, close to the ground, allowing new roots and vertical shoots to develop [44,51].

In the case of *P. tarapacana*, our results show that the dwarf trees (Dt) life form had a lower vitality than the arborescents (Ar). This could be, as in the case of *Pinus aristata*, due to the exposure of the trunk in this tree life form to wind-induced desiccation and wind-driven cambial dieback [54,55]. The tree life form Ar had a greater general vitality, since its trunk is covered with branches. In the case of brousse tigrée (Bt) and Sh, multiple trunks protect one another, generating intermediate values of vitality. This could explain the increase in the proportion of Ar as elevation increases, because of the extreme characteristics of the climate [23].

In *P. tarapacana* forests, brousse tigrée occurs perpendicular to the slope line, where the smallest are down to the slope and the tallest, with a low presence of live leaves, are upslope. The formation of banded vegetation patterns on the slopes responds to a deposition of sediments by the interception of plants [56,57] that affects the properties and structure of the soil, leading to the deterioration of the environment [58] and to mortality in the upper slope vegetation bands. The form and function of different life forms of the alpine plant communities reflect various avoidance, tolerance, or resistance strategies to the interactions of cold temperature, radiation, wind, and desiccation stresses that prevail in the short growing season [59]. In this way, it can be considered that each *P. tarapacana* life form fills a particular niche, so each life form has different adaptations in response to different environmental conditions. The concept of life form as a morphological expression of belonging to a group can help to understand the functioning of this species.

### 3.3. Influence of Topographic, Climatic, Life Zones and Human Use Factors in the Frequency of Life Forms

A life form is the morphological outcome of a number of selection pressures, both abiotic (e.g., climate and elevation) and biotic (e.g., competitive interactions and human uses). It is a structural and functional compromise that allows for the optimization of cost–benefit relationships [60]. In *P. tarapacana*, Hoch and Körner [21] found that the greater the abundance of shrubby forms, the higher the elevation. However, not only elevation influences the general behavior of tree life forms, but also slope, temperature, life zone and resource use, as observed in this study.

Elevation represents a complex combination of climatic variables to which species have to adjust and has been considered an important environmental factor affecting community structure and organization [61,62]. Two variables closely related to elevation are temperature and precipitation [63] and this is the best example of a complex gradient where resources change [12]. Although elevation is the most important variable explaining differences in tree shape, this factor indisputably exercises an indirect influence through interactions with temperature, humidity and topography [17,64]. The development of *P. tarapacana* in higher elevation areas is limited by temperature drops [21,65], where forms that protect the cambium dieback are necessary. According to an ecophysiological study in the Chilean Altiplano, the photosynthetic processes and carbon assimilation of this species are well adapted to withstand cold temperatures [66], which allows them to thrive at elevations higher than 4700 m a.s.l. This was observed in this study, highlighting the differences found in the frequencies of Ar, Dt and Sh with changes in temperature as well as the different life zones [67].

The structure of the studied areas presented smaller trees and a higher crown cover when the elevation decreased [23], where there was a higher proportion of Sh and Bt. This response could be the result of the combined effect of the decrease in temperature and the increase in precipitation as elevation increases [68], increasing the proportion of the Ar form over the others.

Moderate to steep slopes are environments where this species presents its greatest development [19,23], because these landscapes provide safe and suitable sites for seed survival, germination and development of individuals, provided by, among other factors, the higher soil moisture and proportion of daytime with shade offered by rocks that reduce soil evaporation, as well as the smaller temperature fluctuations moderating competition for water with herbaceous vegetation [69]. We observed that the frequency of Ar decreases with the slope, while the Bt form increases its frequency considerably, mainly due to the instability of the soil, which produces the banded shape for this species, and the damages caused by this movement of gravel and soil reduces the frequency of Ar forms. We observed that the frequency of single-trunk forms (Ar and Dt) increases in alpine life zones (TAMT and TAWT), while in subalpine zones (TSDS and TSMF), the Sh form increases in proportion. This could be due to alpine areas presenting low temperatures and intermediate precipitation values, optimal factors for the development of this species [23], with the Sh form being more adapted to higher temperatures and low rainfall [26,37,70]. We observed the largest number of Sh and Bt specimens farther from the towns. This could be related to human uses and access to these areas, since individuals at lower elevations are the most accessible to the local communities, who select the Dt form that has larger dimensions [23,24] for fuel and construction purposes [35]. This is due to the fact that the trunks of *P. tarapacana* in the Argentinean highlands were used as beams for the construction of the roofs of houses, requiring trunks of at least 2–3 m in length [35]. However, the high durability of the wood of this species allows people to not need to replace the cut pieces frequently, generating a lower impact on forests. Likewise, in lower elevation regions (below 3500 m a.s.l.), there are forests of other tree species (*Polylepis tomentella* Wedd. and *Strombocarpa ferox* (Griseb.) C.E. Hughes and G.P. Lewis) that are greatly preferred compared with *P. tarapacana*, reducing its use in those regions [35].

The results of this study indicate the possible life forms that can be managed sustainably. In the case of multi-trunk life forms (Bt and Sh), it is possible to extract those that have a low vitality or are totally dry for their use for fuel or construction. In this context, the specimen is not completely eliminated, maintaining the conservation of genetic diversity [71]. In addition, it is important to consider the different life forms of *P. tarapacana* when conducting conservation studies and management plans, since each life form occupies a particular environmental situation and conserving these forests without differentiating its life forms is to lose part of its ecological niche.

The results of this investigation are preliminary, due to the short period of research and being concerned with an arboreal species with a very slow growth and long life, and the lack of information about how these life forms are subject to succession. It is necessary to continue with studies on the structure of these forests and their relationship with environmental characteristics, taking into account *P. tarapacana* life forms. Likewise, we suggest to carry out studies that evaluate the combined effect of biotic and abiotic factors on various forms of tree life to identify the important factors that have a major influence on their distribution and biometric characteristics.

## 4. Materials and Methods

### 4.1. Study Species

*Polylepis tarapacana* (*Rosaceae*: *Sanguisorbeae*) is a species that constitute rare, monospecific evergreen forests distributed in the sem-iarid high beds of the western Altiplano from southern Peru to south-western Bolivia, northern Chile and adjacent northwest Argentina (16°–23° S) [72] at 3400–5013 m a.s.l. [18,23,73,74]. This species is adapted to the Altiplano, which is capable of withstanding harsh climatic conditions. Under different life forms, *P. tarapacana* ranges from small shrubs to trees up to 7 m height [75]; however, specimens usually oscillate between 1 to 5 m high [73]. *Polylepis tarapacana* is a unique tree because it lives at a higher elevation compared to any other tree species and comprises the highest elevation tree line on Earth [74].

### 4.2. Study Area

We worked in shrublands and forests of *P. tarapacana* distributed in the Andean cordillera in the Province of Jujuy, Argentina (22°04′–23°40′ SL at 66°46′–65°49′ WL; Figure 4), located in the high peaks of the Argentinean Andes mountain range from 4160 to 4952 m a.s.l. [23]. The climate is cold and dry with strong winds, characterized by a reduced seasonality of temperature, but marked seasonality in precipitation [76,77] with 135 to 165 mm.yr^−1^ concentrated in summer, and with 4.2 to 6.5 °C of annual mean temperature [23]. The vegetation in this area is composed of many species with traits linked to extremely low temperatures, wind and xerophytism [77], and specially with dwarf shrubs and cushion plants [20,78,79].

### 4.3. Classification of Life Forms of P. tarapacana

The considered life forms were arborescents (Ar), dwarf trees (Dt), shrubs (Sh) and brousse tigrée (Bt), where Ar is an intermediate form between trees and shrubs, with a single base and branches that arise from the base of the trunk and along the main axis of the tree. Dwarf trees have a single main stem shorter than 5 m, where the trunk is not covered by branches, and well-developed lateral branches forming a crown. Shrubs are multi-stemmed short woody plants, branching at the ground with vegetative buds to form new shoots. The brousse tigrée tree life form occurs in bands perpendicular to the slope line (Figure 1).

### 4.4. Data Obtained

We worked in 70 *P. tarapacana* shrublands and forests patches throughout the distribution range [23]. The patches were selected according to: (i) homogeneous cover, where the distance between individuals was nearly constant; (ii) accessibility; and (iii) the patch size being >1 ha. In the center of each patch, we established one plot (20 × 50 m) to describe the vegetation structure, maintaining the elevation level. We corrected the areas according to the slope of the terrain using the following formula: Corrected area = Area × cosine (slope in degrees) [80]. In this way, the starting point and the ending point of the plot had the same elevation. We measured all live plants ≥0.20 m height, recording: (i) tree life form of each individual; (ii) diameter at the base (DAB, cm), corresponding to the trunk or the tallest trunk of multi-stem plants (Figure 5); (iii) height (H, cm) of the trunk or tallest trunk of multi-stem trees (Figure 5); (iv) diameter of the maximum axis and of the axis at 90 degrees from the each crown to calculate the area of the tree crown (TC, m^2^) with the ellipse formula and the crown spread ratio (CsR) as the relation between the two measurements; and (v) vitality or health status of each individual to test the influence of wind damage for the different tree life forms into 3 types: (a) low vigor (more than 50 % of the foliage light green, more than 50% of dead branches and more than 50% of dead trunk section), (b) intermediate vigor (less than 50% of light green foliage, less than 50% of dead branches and less than 50% of dead trunk section) and (c) healthy (deep green foliage, no dead branches and no damage in the trunk). The measurements were made in March–April and October–November, coinciding with the periods of less rain and intermediate temperatures.

### 4.5. Environmental Characterization

In all the plots, we registered longitude, latitude and elevation (m a.s.l.) with a global geopositioning device (GPS), slope using a clinometer (°) and the aspect with compass as sine and cosine functions of the north magnetic direction. Sine values ranged from −1 (west) to 1 (east), while cosine values ranged from −1 (south) to 1 (north) [80]. The environmental characteristics of each plot were registered through Holdridge life zones following Derguy et al. [67]: (i) Tropical alpine moist tundra (TAMT), (ii) Tropical alpine wet tundra (TAWT), (iii) Tropical subalpine dry scrub (TSDS) and (iv) Tropical subalpine moist forest (TSMF). The climatic factors, annual mean temperature (AMT) (°C) and annual precipitation (AP) (mm.yr^−1^), for the period of 1970–2000 were obtained from WorldClim [81], the values of the human footprint (HF) were extracted from Lizárraga and Monguillot [70], and the distance to the nearest town (DTT) was obtained using QGIS software.

### 4.6. Data Analysis

We conducted non-parametric Kruskal–Wallis tests to compare forest structure variables (DAB, H, TC, CsR and vitality) for the different life form categories determined for *P. tarapacana*. Additionally, we also performed non-parametric Kruskal–Wallis tests using topographic, climate, geographical and human use variables as the main factors to analyze the frequency of each life form. Differences were determined by comparisons of the means (Conover–Iman test at *p* < 0.05).

To characterize the life forms, we used the maximum relative frequency of the plots in relation to topographic, climatic and human uses, where each plot was categorized into single, double, triple and multiple categories in relation to the abundance of the different life forms. Single categories were considered when a frequency was >70% for a certain tree form (e.g., 74% life form Ar, 12% Dt, 8% Sh, 6% Bt = Category Ar) and double categories when the single categories do not reach 70% but the sum of two higher categories reaches >70% (e.g., 54% life form Ar, 22% Dt, 16% Sh, 8% Bt = Category Ar|Dt). The multiple (M) category is when three or more categories are needed to reach >70% frequency (e.g., 36% life form Ar, 24% Dt, 24% Sh, 16% Bt = Category M). When the categories were represented by less than 5% of the plots (<4 plots), these categories were grouped creating a new category (e.g., Ar|Dt = 3 plots; Ar|Sh = 2 plots: Ar|Dt|Sh = 5 plots). With these categorizations, the means and the standard deviations were calculated for each topographic, climatic and human use. Additionally, the variability of the plots was determined in terms of the simultaneous occurrence of the different forms life forms depending on the topography, climate and human use.

## 5. Conclusions

A life form is the morphological result of selection pressures, both abiotic (e.g., climate and altitude) and biotic (e.g., competitive interactions and human uses). In this study, a new proposal for the classification of *P. tarapacana* life forms was presented, and we demonstrated the influence of elevation, slope, life zone, AMT and DTT in the life form frequency. Each life form of *P. tarapacana* occupies a particular niche, and this concept can help us to understand the ways in which these Altiplano communities function.

In this study, the relative success of each life form was measured in terms of relative frequency, but other measures, such as the biomass of each life form, may be useful and provide additional information. This study showed a general consistency in the changes in life forms of the high-altitude Andean vegetation, and the changes generated by environmental gradients in this species were explained. This consistency provides a framework for broader comparisons with species of the genus *Polylepis* in other parts of South America. These comparisons will provide information on the distribution of these and other life forms and may help us to understand the mechanisms that determine the structure of these high-elevation forest communities and beyond.

## Figures and Tables

**Figure 1 plants-12-01806-f001:**
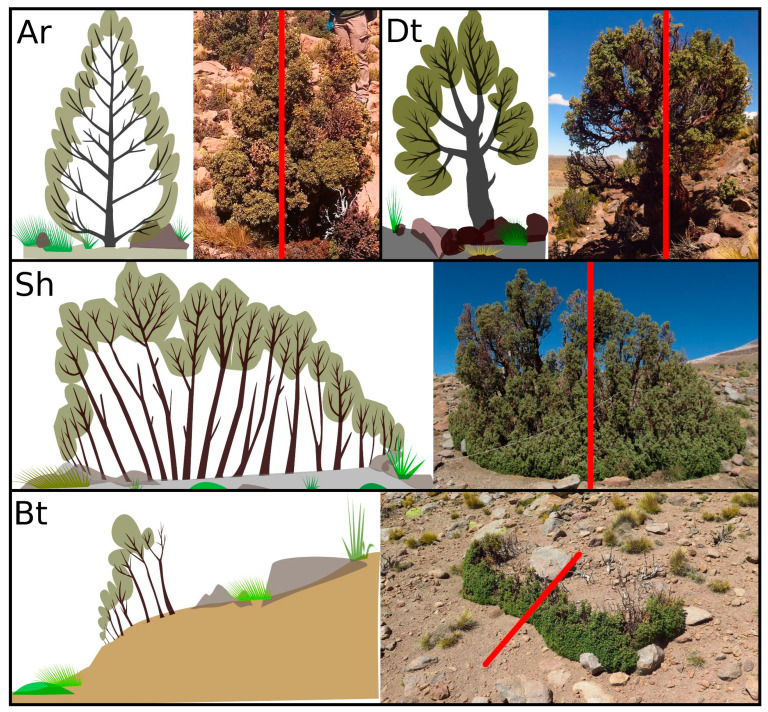
Classification of life forms in *P. tarapacana.* Ar: Arborescent; Dt: Dwarf tree; Sh: Shrubs; Bt: Brousse tigrée. The red line in the photo indicates the vertical cut that is observed in the graph on the left.

**Figure 2 plants-12-01806-f002:**
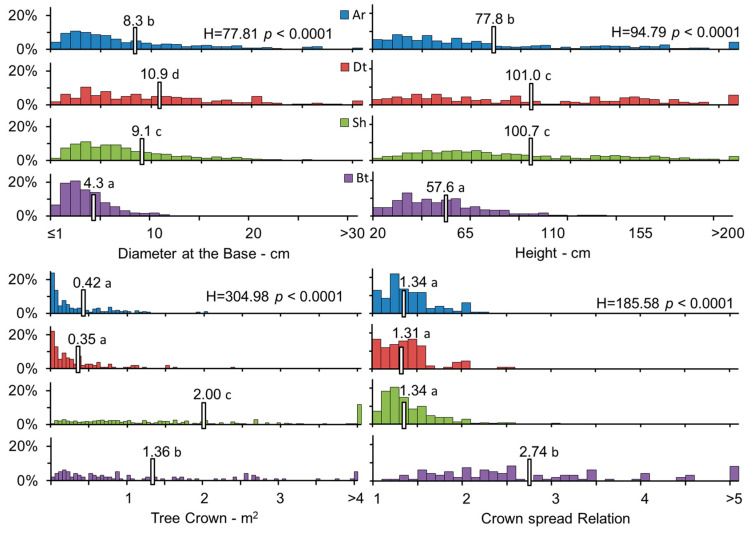
Kruskal–Wallis test for the diameter, height, tree crown and crown spread ratio of *P. tarapacana* life forms. Different letters indicate significant differences (*p* < 0.05) by Conover–Iman test.

**Figure 3 plants-12-01806-f003:**
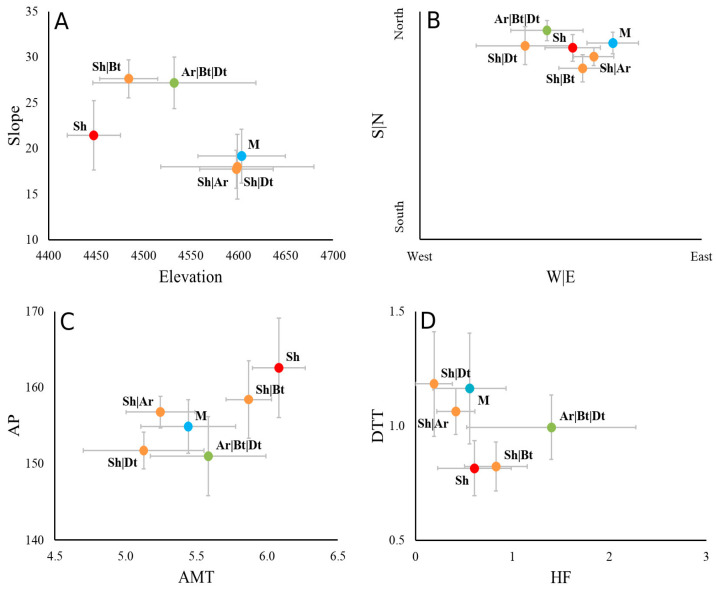
Topographic, climatic and human use factors classified by the frequency of life forms. Relationship among topographic variables ((**A**) Elevation and slope, (**B**) Aspects), climatic ((**C**) temperature and precipitation) and human use ((**D**) human footprint and distance to towns). Bars indicate the standard deviation of each axis. Elevation in m a.s.l.; Slope in degree; S|N: North Aspect; W|E: East aspect. The aspects factors were calculated as sine and cosine functions, where sine values range from −1 (west) to 1 (east), while cosine values range from −1 (south) to 1 (north). AMT: Annual mean temperature in °C; AP: Annual precipitation in mm.yr^−1^; HF: Human footprint; DTT: Distance to towns in km. Ar: Arborescents; Sh: Shrubs; Dt: Dwarf trees; Bt: Brousse tigrée; M: Multiple forms.

**Figure 4 plants-12-01806-f004:**
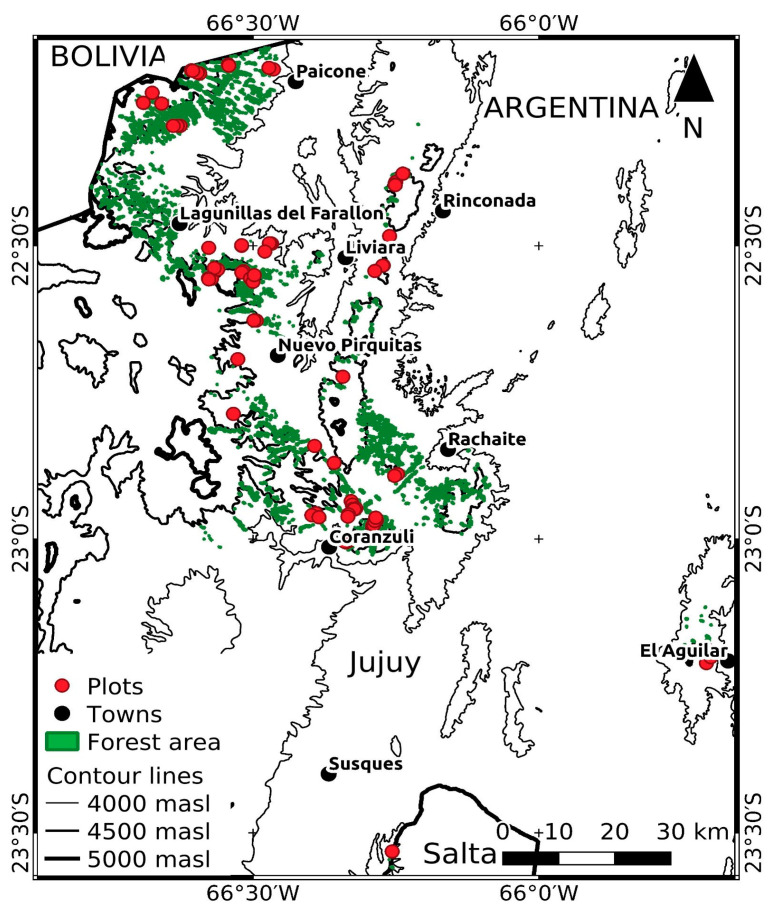
Distribution of *Polylepis tarapacana* forests (green) in the study area (Altiplano, Argentina), showing plots (red circles), towns (black circles) and contour lines, at 4000 m a.s.l. (narrow line), 4500 m a.s.l. (medium line) and 5000 m a.s.l. (tick line). Modified from López et al. [23].

**Figure 5 plants-12-01806-f005:**
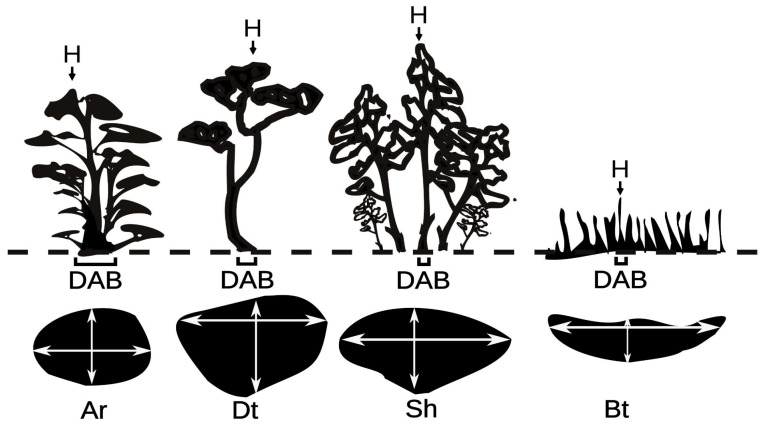
Determination of the height, diameter at the base, crown diameter of the maximum axis and of the axis at 90 degrees for each life forms in *P. tarapacana*. H: height; DAB: diameter at the base of the tree; Ar: Arborescent; Dt: Dwarf tree; Sh: Shrubs; Bt: Brousse tigrée.

**Table 1 plants-12-01806-t001:** Kruskal–Wallis test for the relative frequency of *Polylepis tarapacana* life forms compared with topographic factors (elevation in m a.s.l., slope in degrees and two aspects). Different letters indicate significant differences (*p* < 0.05) by the Conover–Iman test.

Variable	Range	n	Ar	Dt	Sh	Bt
Elevation	<4400	14	17 a	5 a	59 b	19 b
	4400–4500	17	17 a	8 ab	51 ab	24 b
	4500–4600	19	26 ab	10 abc	53 ab	11 a
	4600–4700	6	30 b	16 c	39 a	14 ab
	>4700	14	35 b	13 bc	40 a	12 a
	*p*		0.0082	0.0349	0.0123	0.0017
	H_Test_		13.7	10.2	12.7	17.1
Slope	<15	18	30 b	11	55	8 a
	15–25	25	27 ab	11	48	14 a
	>25	27	18 a	7	50	24 b
	*p*		0.0268	0.2320	0.5501	0.0002
	H_Test_		7.2	2.8	1.2	17.2
NA	N	27	22	8	52	18
	Rest	43	25	10	49	15
	*p*		0.3250	0.3049	0.5542	0.2000
	H_Test_		0.9	1.0	0.3	2.7
EA	W	13	28	10	50	12
	Rest	34	22	10	51	17
	E	23	24	9	50	17
	*p*		0.6889	0.9677	0.9456	0.5978
	H_Test_		0.7	0.1	1.1	1.0

Elevation in m a.s.l.; slope in degrees; NA: North aspect (N: from 315° to 45°); EA: East aspect (E: from 45° to 135°, W: from 225° to 315°). The aspects were calculated as sine and cosine functions, where sine values range from −1 (west) to 1 (east), while cosine values range from −1 (south) to 1 (north). Ar: Arborescents; Sh: Shrubs; Dt: Dwarf trees; Bt: Brousse tigrée. Different letters indicate significant differences determined by comparisons of means (Conover–Iman test, *p* < 0.05).

**Table 2 plants-12-01806-t002:** Kruskal–Wallis of the frequency of *Polylepis tarapacana* life forms compared with climatic (temperature and precipitation), geographical (life zones), human footprint and distance to towns factors.

Variable	Range	n	Ar	Dt	Sh	Bt
Life Zone	TAMT	8	42 b	17 b	36 a	5 a
	TAWT	2	39 ab	18 ab	39 ab	5 ab
	TSDS	43	18 a	8 a	52 ab	22 b
	TSMF	17	29 b	8 a	55 b	8 a
	*p*		0.0006	0.0446	0.0426	0.0001
	H_Test_		17.3	7.9	8.2	21.8
AMT	<5	15	34 b	15 b	39 a	11
	5–6	33	22 a	9 a	52 b	17
	>6	22	20 a	7 a	55 b	18
	*p*		0.0340	0.0232	0.0199	0.0629
	H_Test_		6.8	7.5	7.8	5.5
AP	<150	26	24	9	50	17
	>150	44	24	10	51	16
	*p*		0.9399	0.6171	0.9890	0.3618
	H_Test_		<0.01	0.3	<0.01	0.9
HF	0	53	25	10	50	15
	>0	17	20	8	50	21
	*p*		0.2758	0.5673	0.9891	0.1700
	H_Test_		1.2	0.3	<0.01	1.9
DTT	<5	13	14 a	6	58	23 b
	5–10	32	23 a	9	51	17 ab
	>10	25	31 b	11	45	12 a
	*p*		0.0039	0.2661	0.1225	0.0436
	H_Test_		11.0	2.6	4.2	6.2

Life zones: Tropical alpine moist tundra (TAMT), Tropical alpine wet tundra (TAWT), Tropical subalpine dry scrub (TSDS) and Tropical subalpine moist forest (TSMF). AMT: Annual mean temperature in °C; AP: Annual precipitation in mm.yr^−1^; HF: Human footprint; DTT: Distance to towns in km. A: Arborescents; Sh: Shrubs; Dt: Dwarf trees; Bt: Brousse tigrée. Different letters indicate significant differences determined by comparisons of means (Conover–Iman test, *p* < 0.05).

## Data Availability

Not applicable.

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
