# Peer review of "Characterization of Polylepis tarapacana Life Forms in the Highest-Elevation Altiplano in South America: Influence of the Topography, Climate and Human Uses"

_plants, 2023, doi:10.3390/plants12091806_

Round 1
Reviewer 1 Report
This manuscript describes an analysis of the effects of topographic, climatic, geographic and human-use factors on four different life forms of Polylepis tarapacana in Argentina. A large sample size was used to quantify the biometry of life forms and evaluate the distribution of life forms with respect to environmental factors. Although the inferred relationships between life forms and environmental factors are not surprising from an ecological perspective, there is some value to documenting these relationships for the distribution of P. tarapacana in this region, especially given that this species has the highest treeline in the world.
The manuscript has clear objectives, straightforward field methods, and a robust analytical approach. Inferences are generally logical and well-supported by the data. I found no major technical issues. Here are a few minor points that should be addressed:
40ff. Somewhere in the Introduction, it should be mentioned that P. tarapacana is listed as “near threatened” by the IUCN.
362-364. “Safe and suitable sites for regeneration” is vague. What does this mean specifically in terms of biophysical conditions? It is likely that P. tarapacana exists here not just because it can germinate but because it is tolerant of the conditions and outcompetes other plants.
490ff. Some discussion is needed on the value of this study for the conservation and management of P. tarapacana, especially given that it is listed as “near threatened” by IUCN.
Breaking up the long paragraphs into smaller paragraphs would improve the readability of the manuscript. It would help to reduce wordiness in the text throughout the manuscript.
The English writing is generally good, although there are minor errors throughout the manuscript. These could be resolved by a technical editor.
The English writing is generally good, although there are minor errors throughout the manuscript. These could be resolved by a technical editor.
Reviewer 2 Report
A life form is the morphological result of a series of selection pressures, both abiotic and biotic. Thus, it reflects various strategies of plant resistance to factors such as: low temperature, radiation, wind, drought, etc. In the conditions of increasingly frequent extreme weather and climate phenomena, understanding the mechanisms of plant adaptation under the influence of various environmental factors is very important. For this reason, the scientific article submitted for review concerns an important issue, which is the identification of internal regulatory mechanisms of plant communities under the influence of many natural and economic factors.
The experiment was fairly well described. However, in the methodological part, the authors did not fully explain the adopted methodological solutions. They didn't write:
· What determined the number of 70 research plots and the plot size (1000 m2) ?
· How was the research area (20x50 m) located throughout its range? Has the exposure of the slope in its location (along the long side) been taken into account? Or maybe the location of the plots was determined by the shape of the forest patch?
· On what basis (perhaps the results of other studies?) the range of altitudinal zones with a span of 100 m was assumed?. Did it depend on the number of research plots at different heights above sea level?
· In the chapter "4. Materials and methods" it is not written when (in what months) the measurements were made. Is this important when assessing the vitality or health of each tree?
· Based on what studies was it assumed that the distance from the city will affect the life form of trees?
The main aim of the work was to show what biometric features characteristic of trees represent different life forms of trees and how the life forms of trees change with topography? . Meanwhile, in the study submitted for review, the parameters of trees in various forms of life are given only jointly, i.e. for all measured trees. This way of analyzing and presenting the results means that the biometric features of trees do not show statistically significant differences in different life forms of trees due to the very high variability. Therefore, the results should be supplemented with a list of biometric characteristics of trees of various life forms, taking into account: Holdridge life zones, altitude above sea level and exposure.
It should be emphasized in the article that these are preliminary research results, because: we do not know whether tree life forms are not subject to succession, i.e. they do not pass from one tree life form to another or are not the result of variability. The obtained results, due to the short period of research, do not provide grounds for creating a final classification of life forms. Only long-term observations covering the entire development cycle of the tree give reliable results.
In addition, I propose that further research and scientific work should also take into account the combined effect of biotic and abiotic factors on various forms of tree life. Additive models or multiple regression models can be used for this purpose. These models are likely to identify important factors that have a major influence on the distribution of different tree life forms and the biometric characteristics of trees in different tree life forms.
Line 4 - it should be the climate (data obtained from WorldClim do not allow for the assessment of microclimatic parameters)
Line 134 - (distance to towns) standard deviation ± 4795 km!!!?
Overall, however, the results were well presented and the whole article is very well done
